# A Novel Four-Gene Score to Predict Pathologically Complete (R0) Resection and Survival in Pancreatic Cancer

**DOI:** 10.3390/cancers12123635

**Published:** 2020-12-04

**Authors:** Masanori Oshi, Yoshihisa Tokumaru, Ankit Patel, Li Yan, Ryusei Matsuyama, Itaru Endo, Matthew H.G. Katz, Kazuaki Takabe

**Affiliations:** 1Department of Surgical Oncology, Roswell Park Comprehensive Cancer Center, Buffalo, NY 14263, USA; masa1101oshi@gmail.com (M.O.); Yoshihisa.Tokumaru@roswellpark.org (Y.T.); Ankit.Patel@RoswellPark.org (A.P.); 2Department of Gastroenterological Surgery, Yokohama City University Graduate School of Medicine, Yokohama 236-0004, Japan; ryusei@yokohama-cu.ac.jp (R.M.); endoit@yokohama-cu.ac.jp (I.E.); 3Department of Surgical Oncology, Graduate School of Medicine, Gifu University, 1-1 Yanagido, Gifu 501-1194, Japan; 4Department of Biostatistics and Bioinformatics, Roswell Park Comprehensive Cancer Center, Buffalo, NY 14263, USA; li.yan@roswellpark.org; 5Department of Surgical Oncology, The University of Texas MD Anderson Cancer Center, Houston, TX 77030, USA; mhgkatz@mdanderson.org; 6Division of Digestive and General Surgery, Niigata University Graduate School of Medical and Dental Sciences, Niigata 951-8520, Japan; 7Department of Breast Surgery, Fukushima Medical University School of Medicine, Fukushima 960-1295, Japan; 8Department of Surgery, Jacobs School of Medicine and Biomedical Sciences, State University of New York, Buffalo, NY 14263, USA; 9Department of Breast Surgery and Oncology, Tokyo Medical University, Tokyo 160-8402, Japan

**Keywords:** alteration, biomarker, gene set, immune cell, pathway analysis, pancreatic cancer, proliferation, resection, treatment response, tumor gene expression, four-gene score

## Abstract

**Simple Summary:**

A biomarker to predict survival is a critical need in pancreatic cancer treatment. We hypothesized that a four-gene score, which was previously reported to reflect cell proliferation, can be used as a predictive biomarker for pancreatic cancer. A total of 954 patients were analyzed for both discovery and validation of the four-gene score from publicly available datasets for pancreatic cancer, in order to investigate the relationship between the score and clinical features of pancreatic cancer, such as metastasis, cancer aggressiveness, immune cell infiltration, patient survival, and resectability. We found that the score correlated with clinical aggressiveness in pancreatic cancer, and did so to a higher degree compared to breast cancer cohorts. We found that the four-gene score identified poor survival in pancreatic cancer, and has potential as a predictive biomarker of treatment response in metastatic pancreatic cancer, as well completion of a pathologically complete (R0) resection. Appropriately utilized, the four-gene score could be a valuable prognostic and predictive tool for pancreatic cancer in the future.

**Abstract:**

Pathologically complete (R0) resection is essential for prolonged survival in pancreatic cancer. Survival depends not only on surgical technique, but also on cancer biology. A biomarker to predict survival is a critical need in pancreatic treatment. We hypothesized that this 4-gene score, which was reported to reflect cell proliferation, is a translatable predictive biomarker for pancreatic cancer. A total of 954 pancreatic cancer patients from multiple cohorts were analyzed and validated. Pancreatic cancer had the 10th highest median score of 32 cancers in The Cancer Genome Atlas (TCGA) cohort. The four-gene score significantly correlated with pathological grade and *MKI67* expression. The high four-gene score enriched cell proliferation-related and cancer aggressiveness-related gene sets. The high score was associated with activation of *KRAS*, *p53*, transforming growth factor (TGF)-β, and E2F pathways, and with high alteration rate of *KRAS* and *CDKN2A* genes. The high score was also significantly associated with reduced CD8^+^ T cell infiltration of tumors, but with high levels of interferon-γ and cytolytic activity in tumors. The four-gene score correlated with the area under the curve of irinotecan and sorafenib in primary pancreatic cancer, and with paclitaxel and doxorubicin in metastatic pancreatic cancer. The high four-gene score was associated with significantly fewer R0 resections and worse survival. The novelty of the study is in the application of the four-gene score to pancreatic cancer, rather than the bioinformatics technique itself. Future analyses of inoperable lesions are expected to clarify the utility of our score as a predictive biomarker of systemic treatments.

## 1. Introduction

Pancreatic cancer is one of the most aggressive malignancies, with a five-year survival of less than 10% [1]. The incidence of pancreatic cancer has increased over the past few decades, and is expected to increase further. In many cancer types, clinical decisions are made based on histopathological criteria; however, this is not enough to manage pancreatic cancer [2]. Recent identification of borderline resectability has allowed pancreatic surgeons to more appropriately select patients for complete curative resection with microscopic tumor clearance (R0), which is known to significantly improve median survival [3]. On the other hand, pancreatic cancer resections that resulted in positive margins (R1 and R2 (R1/2)) are thought to be influenced by aggressive tumor biology and invasiveness of the cancer, outside of surgical technique alone [4,5,6].

In recent years, there has been a trend to administering neoadjuvant therapy followed by resection. The benefits of a neoadjuvant treatment include the greater potential of achieving complete resection with negative surgical margins, as well as potentially spare patients with rapidly advancing cancer from the morbidity of major surgery. It may also decrease systemic recurrence [7,8]. However, these advances in surgery and systemic chemotherapy have not significantly improved the prognosis of pancreatic cancer [9]. Unfortunately, the majority of patients present with metastatic disease at the time of diagnosis, and only 20% present with surgically resectable disease. Systemic chemotherapy remains the primary treatment option for the majority of patients. However, there is lack of consensus as to which therapy offers the best efficacy and the quality of life for patients. Given the insidious clinical presentation and poor treatment response of pancreatic cancer, there is an urgent and critical need to discover a predictive biomarker to assist with appropriate patient selection in the treatment of pancreatic cancer.

We have previously reported a novel four-gene score as a prognostic and predictive biomarker that reflect cell proliferation for breast cancer and its metastasis [10]. The score is derived from tumor expression of docking protein-4 (*DOK4*), holocytochrome c synthase (*HCCS*), placental growth factor (*PGF*), and SHC SH2 domain-binding protein-1 (*SHCBP1*) genes, which were identified based on differential mRNA expression analysis of a human breast cell line and its metastatic variant cells, and with the clinical outcome data of breast cancer patient cohorts. One interesting feature of the score is that it correlates with increased proliferation-related gene sets. Aberrant cell proliferation, arguably the most heavily studied hallmark of cancer, is known to play a major role in aggressiveness of pancreatic cancer [11]. Thus, we hypothesized that the four-gene score, which reflects tumor cell proliferation, is both a prognostic and predictive biomarker for pancreatic cancer. To test this hypothesis, we analyzed 954 human pancreatic cancer patients from multiple cohorts for both discovery and validation of the four-gene score from publicly available datasets for pancreatic cancer, and examined whether it reflects cell proliferation and associates with gene alteration, immune cell infiltration, pathologically complete resection (R0), and survival of pancreatic cancer patients.

## 2. Results

### 2.1. The Four-Gene Score Is Elevated in Pancreatic Cancer Tumors

As we previously reported, the four-gene score has reflected cell proliferation in breast cancer by measuring gene expressions of *DOK4*, *HCCS*, *SHCBP1*, and *PGF* [10]; it was of interest to investigate the distribution of the score in other types of cancer in The Cancer Genome Atlas (TCGA) project [12]. Figure 1 demonstrates the four-gene score levels of different cancer types in TCGA. The four-gene score was most elevated in head and neck squamous cell cancer (HNSC), followed by rectal and colon cancer (READ and COAD, respectively). Pancreatic cancer had the 10th highest median level for the four-gene score among the 32 cancer types in TCGA.

### 2.2. The Four-Gene Score Correlates with Pathological Grade and MKI67 Expression, and Was Elevated in Metastatic Pancreatic Cancer

Following our findings that the four-gene score reflects cell proliferation in breast cancer, we hypothesized that a high score is associated with aggressiveness in pancreatic cancer. To test this hypothesis, we first studied the association of the four-gene score with clinical aggressiveness of pancreatic cancer in the TCGA and GSE62452 [13] cohorts, which contain clinical information. Although the four-gene score was not associated with American Joint Committee on Cancer (AJCC) stage (*p* = 0.489 and 0.126, respectively), it was associated with higher pathological grade in both cohorts (Figure 2A; *p* < 0.001, and *p* = 0.034, respectively). Next, we studied the correlation of the score with *MKI67* gene expression, one of the most commonly used cell proliferation markers, in three cohorts: TCGA, GSE62452, and GSE57495 [14]. The four-gene score correlated strongly with *MKI67* gene expression in these cohorts (Figure 2B; Spearman’s rank correlation coefficient [*r*] = 0.610, 0.761, and 0.690, respectively; all *p* < 0.01). Finally, we compared the four-gene score of the primary and metastatic pancreatic cancer using GSE71729 [15] and GSE34153 [16] cohorts, which included metastatic tumors. The four-gene score was the highest in metastatic tumors compared with primary pancreatic cancer and normal pancreas tissue, in that order, in the GSE71729 (Figure 2C; *p* < 0.001), and was also higher than primary pancreatic cancer in the GSE34153 cohort (Figure 2C; *p* < 0.001). These findings suggest that the four-gene score in pancreatic cancer was associated with tumor aggressiveness.

### 2.3. Tumors with High Four-Gene Score Have Enriched Expression of Cell Proiferation-Related and Pro-Cancerous Gene Sets

Given the results of Figure 2, we expected that the four-gene score reflects the cell proliferation of pancreatic cancer. In order to elucidate the mechanism, TCGA, GSE62452, and GSE57495 cohorts were divided into low and high groups of patients based on the intra-cohort median for the four-gene score, and gene set enrichment analyses (GSEA) with MSigDB Hallmark gene sets were conducted (Figure 3). Tumor gene expression for the high four-gene score group was enriched for all cell proliferation-related gene sets (E2F target, G2M checkpoint, MYC targets v1, MYC targets v2, and mitotic spindle) consistently in all three cohorts (Figure 3). Furthermore, the high four-gene score group also had enrichment for cancer aggressiveness-related gene sets, such as MTORC1 signaling, PI3K/AKT/MTOR signaling, DNA repair, unfolded protein response, hypoxia, and notch signaling (Appendix A). These results suggest that the four-gene score reflects not only cell proliferation, but also other cancer aggressiveness in pancreatic cancer.

### 2.4. Pancreatic Cancer with High Four-Gene Sore Have Higher KRAS and CDKN2A Gene Alterations, and Enhanced Signaling of KRAS, p53, TGF-β, and E2F Target Pathways

*KRAS*, *TP53*, *SMAD4*, and *CDKN2A* genes are known to be frequently altered, and are associated with poor prognosis in pancreatic cancer [17]. To this end, we hypothesized that the high four-gene score is associated with high rate of gene alteration, and with enhanced signaling of the pathways in which these genes are involved. The activation of each signaling pathways were quantified using gene set variation analysis (GSVA), as we previously reported [18,19,20,21,22]. We found that the high four-gene score group was significantly associated with a higher percentage of alteration in *KRAS* and *CDKN2A*, but not with *TP53* or *SMAD4* in the TCGA cohort (Figure 4A; *p* = 0.006, *p* = 0.044, *p* = 0.059, and *p* = 1.00, respectively). The tumors with a four-gene scores were significantly associated with less KRAS down-signaling, elevated P53 pathways, transforming growth factor (TGF)-β signaling, and E2F targets consistently in all three cohorts, which suggest that all four pathways are enhanced in those tumors (Figure 4B).

### 2.5. Tumors with High Four-Gene Score Have High Levels of Interferon -Γ Response and Cytolytic Activity

Since cell proliferation and the aggressiveness of cancer can be affected by the tumor immune microenvironment [18,20,23], we expected that high-four-gene-score pancreatic cancer to have a characteristic immune cell infiltration profile. Utilizing the xCell algorithm [24], we found that pancreatic cancer with a high four-gene score was significantly associated with a lower fraction of CD8^+^ T cells consistently in the TCGA, GSE62452, and GSE57495 cohorts (Figure 5A; *p* = 0.037 and 0.002, and *p* < 0.001, respectively). Other type of immune cells (CD4^+^ memory T cells, M1 and M2 macrophages, and regulatory T cells) were not associated with the four-gene score in these cohorts. Interestingly, the Hallmark interferon (IFN)-γ response gene set was significantly enriched in high-four-gene-score tumors by GSEA (Figure 5B; NES = 1.55 (FDR = 0.05), 1.18 (FDR = 0.32), 1.37 (FDR = 0.18), respectively). This result was consistent with the high-four-gene score pancreatic cancer being significantly associated with enhanced cytolytic activity score (CYT) and IFN-γ response score in the TCGA cohort (Figure 5C; *p* = 0.014, and 0.004, respectively). These results suggest that high-four-gene score pancreatic cancer is associated with favorable immune microenvironment.

### 2.6. The Four-Gene Score Level Associates with Drug Sensitivity to Sorafenib in Primary Pancreatic Cancer, and Paclitaxel and Doxorubicin in Metastatic Pancreatic Cancer

Since the four-gene score is associated with cell proliferation, which, in turn, is known to correlate with the efficacy of chemotherapy, we hypothesized that the four-gene score predicts response to chemotherapy in pancreatic cancer. To test this hypotheses, we investigated the correlation of the four-gene score with response to chemotherapy as the level of the area under the curve (AUC) of drug sensitivity of gemcitabine, paclitaxel, doxorubicin, irinotecan 5-fluorouracil, oxaliplatin, sunitinib, and sorafenib, using multiple cell lines derived from either primary or metastatic pancreatic cancer (Appendix A). Interestingly, the four-gene score correlated with the AUC of irinotecan and sorafenib in primary pancreatic cancer cell lines (Figure 6; *r* = 0.587 (*p* = 0.03) and *r* = 0.648 (*p* = 0.02), respectively), whereas it correlated with the AUC of paclitaxel and doxorubicin in metastatic pancreatic cancer cell lines (*r* = 0.582 (*p* = 0.02) and *r* = 0.624 (*p* < 0.01), respectively). These results suggest that the four-gene score may have a role in identifying patients who respond to those drugs that are currently not widely used for pancreatic cancer, and further studies using animal models and prospective cohort are warranted.

### 2.7. Pancreatic Cancer with a High Four-Gene Score Was Significantly Associated with Worse Survival and Pathologically Margin-Positive Resection

Given that the four-gene score was associated with pancreatic cancer aggressiveness, we hypothesized that high-four-gene-score pancreatic cancer is associated with worse survival. We found that the high-four-gene-score group was significantly associated with worse disease-free survival (DFS), disease-specific survival (DSS), and progression-free survival (PFS) in the TCGA cohort (Figure 7A upper; *p* = 0.026, 0.027, and 0.019, respectively). Furthermore, the high-score group was significantly and consistently associated with worse overall survival (OS) in the TCGA, GSE62452, and GSE57495 cohorts (Figure 7A lower; *p* = 0.044, 0.048, and 0.022, respectively).

Margin positivity (cancer cells present at the resected specimen margin) after resection is speculated to be not only due to surgical technique, but also cancer biological aggressiveness [4,5,6]. Margin-positive status (R1 or R2 (R1/2)) is known to be one of the risk factors for worse survival. To this end, we hypothesized that high-four-gene-score pancreatic cancer is more likely to have positive margin (R1 or R2). To test this hypothesis, low- and high-four-gene-score tumors were compared by an achievement rate of R0 (pathologically margin-negative) resection. We found that high-four-gene-score group was significantly associated with lower R0 resection rate compared with the low-score group in the TCGA cohort (Figure 7B, *p* = 0.013). Furthermore, although there was no survival difference by four-gene score in patients who achieved R0 after resection, the high-four-gene score tumors were significantly associated with worse DSS in patients who had R1 or R2 (R1/2) resection in the TCGA cohort (Figure 7C, *p* = 0.477 and *p* = 0.027, respectively). These results implicate that the four-gene score can be a predictive biomarker for R0 resection and a prognostic biomarker for pancreatic cancer, particularly in aggressive R1/2, margin-positive pancreatic cancer.

## 3. Discussion

In this study, a total of 954 human pancreatic cancer patients from a discovery cohort (TCGA) and three completely independent validation cohorts (GSE57495, GSE62452, and GSE71729) were analyzed for the clinical relevance of the four-gene score to metastasis, aggressiveness, immune cell infiltration, patient survival, and resectability of pancreatic cancer. The four-gene score correlation in pancreatic cancer was higher than in breast cancer cohorts, specifically in the clinical aggressive parameters, which included pathological grade and *MKI67* expression. Furthermore, the score in metastatic tumor cohorts was higher than in the primary pancreatic cancer cohorts. These findings were mirrored in the gene set analysis, in which the high-four-gene-score tumors enriched cell proliferation-related gene sets, such as E2F targets, G2M checkpoint, MYC targets v1, MYC targets v2, and the mitotic spindle, as well as metastasis-related gene sets. Additionally, a high four-gene score was also associated with increased mutation rates in *KRAS* and *CDKN2A* genes, as well as enhancement of *KRAS*, *CDKN2A*, *p53*, TGF-β, and E2F pathways. Furthermore, a high four-gene score tumor had less infiltration of CD8^+^ T cells, but was associated with high CYT and IFN-γ scores, which reflects overall cancer immunity. Interestingly, the high score was correlated with the level of drug sensitivity AUC of irinotecan and sorafenib in primary cancer, and with that of paclitaxel and doxorubicin in metastatic pancreatic cancer cell lines. Finally, a high four-gene score was significantly associated with worse clinical outcomes, a low rate of R0 resection, and worse survival after R1/2 resection. The novelty of the study is in its application of the four-gene score to pancreatic cancer, though not in the bioinformatics techniques itself, nor the discovery of a novel function of the four genes. Although there is no doubt in the importance of elucidating the function and mechanism of genes using in vitro and in vivo experimental settings, no system can perfectly model human cancer and its tumor microenvironment. Cell culture systems lack tumor microenvironments all together, and in vivo systems often lack human cancer cells, heterogeneity, or intact immune cells. The results obtained using these artificial systems always leave a fundamental question as to whether the result applies to human cancer. The aim of the current study was not to report a new function of these four genes, but rather to highlight the translation of utility from breast cancer to pancreatic cancer in its ability to predict complete pathology resection (R0) and survival. This is in agreement with previous reports that also indicate that findings based on single gene expression often suffer limited reproducibility, and the interpretation of their biological meaning can be challenging. Cancer biology is often dictated by the concerted functions of multiple genes [25], and gene expression scoring that captures multiple genes allows consolidation of the complex system into single-gene scoring [26]. A score with multiple genes can take into account such coordination of genes, reduce model complexity, and increase the explanatory power of prediction models [27,28,29].

Among the four genes that constitute the score, placental growth factor (*PGF*) alone has been reported in association with pancreatic cancer [30,31]. *PGF*, which is a member of the vascular endothelial growth factor (*VEGF*) family, is involved in several diseases associated with pathological angiogenesis [32,33]. Sabbaghian et. al. reported that an elevation of *PGF* in plasma may be related to neovascularization of pancreatic cancer, and suggested it as a possible target for anti-angiogenesis therapy [34]. However, the association between *PGF* expression in pancreatic cancer tumors and prognosis has never been reported.

SHC SH2 domain-binding protein 1 (*SHCBP1*) is an important connexin on the SH2 domain of the SHC protein, and its functional role has not been clearly established [35]. The mRNA and protein of the *SHCBP1* gene are expressed in proliferating cells, including cancer cells, but are not expressed in stable cells, such as skeletal muscle and cardiomyocytes [36]. *SHCBP1* is an important intracellular signaling pathway protein, which has been demonstrated to mediate multiple signaling pathways, such as *RAS* and *PI3K/AKT*, and has a role in regulating the cell cycle and promoting cell migration and invasion [37]. However, the exact role of *SHCBP1* in pancreatic cancer stays unclear.

Docking protein-4 (*DOK4*, also called downstream of tyrosine kinase 4), or insulin receptor substrate-5 (*IRS5*), acts as an anchor for c-Src kinase, inhibits tyrosine kinase signaling and can activate *MAPK* [38,39]. The expression of *DOK4* gene was reported to regulate chromatin remodeling in non-small-cell lung cancer [40]. *DOK4* has been reported as a potential biomarker for prognostic outcomes in several cancers [41,42,43], but not in pancreatic cancer.

Holocytochrome c synthase (*HCCS*) effects cellular levels of cytochrome C, impacting mitochondrial physiology and cell death [44]. The disease, microphthalmia with linear skin defects (MLS), is due to mutations in the *HCCS* gene [45,46]. There has been no publication that reports an association of *HCCS* with any cancer.

Recently, it has become clear that the tumor immune microenvironment (TIME) plays a significant role in cancer progression. The presence of tumor-infiltrating lymphocytes (TILs) and the composition of the TIME predict prognosis and treatment response. Previously, we found that anti-cancer immunity counterbalances the biological aggressiveness of high mutation-derived breast cancer, in which there was no survival difference in those tumors [23]. On the other hand, we also found that breast cancer with increased cell proliferation, such as enhanced E2F pathways [18] or G2M cell cycle pathways [20], was associated with elevated anti-cancer immunity, such as an IFN-γ response. However, these cases demonstrated worse survival, likely because extreme cell proliferation overwhelmed the anti-cancer immunity of the immune cells. We found that both IFN-γ response and CYT were elevated in high-four-gene-score pancreatic cancer, which implies that there is anti-cancer immunity in those tumors. Given that a high four-gene score in pancreatic cancer has been associated with worse survival, we speculate that high-four-gene-score tumors are aggressive, proliferative, and easily overwhelm anti-cancer immunity. Pancreatic cancer is often referred to as either an altered or cold immune tumor [47,48] because of the low number of TILs compared with other cancer types, likely allowing anti-cancer immunity to be overwhelmed. In agreement with this notion, CD8^+^ T cells (cytotoxic T cells), the major effector cell in recognizing and killing cancer cells [49] and a favorable prognostic marker in pancreatic cancer patients [50,51,52,53], were less infiltrated in high-four-gene-score pancreatic cancer.

A prognostic biomarker is critical in pancreatic cancer for appropriate patient selection for aggressive treatment. The four-gene score was prognostic for worse survival of pancreatic cancer in multiple cohorts, particularly in R1/2 resection patients. Furthermore, the high-score pancreatic cancer was significantly associated with a high rate of R1/R2 resection. There is increasing evidence that cancer biology is the determining factor in achieving complete curative resection, in addition to surgical techniques [4,5,6]. Our results suggest that the four-gene score has a potential to be used as a predictive biomarker for a successful operation and achieving an R0 resection, by reflecting the aggressiveness of pancreatic cancer.

Systemic chemotherapy has been the primary modality of neoadjuvant treatment for locally advanced unresectable or recurrent pancreatic cancer. Clinical trials with gemcitabine produced only modest clinical benefits and marginally increased survival in patients with advanced pancreatic cancer [54]. For 80% of patients with metastatic pancreatic cancer who tolerate the most effective evidence-based treatment regimen of FOLFIRINOX (folic acid, 5-fluorouracil, irinotecan, and oxaliplatin), the median overall survival time is only 11 months [55]. Although major efforts are underway to effectively target some tumor genes and the tumor microenvironment, targeted therapies in advanced pancreatic cancer do not show significant improvement in survival [56]. Establishing predictive biomarkers that can improve therapeutic efficacy and reduce therapeutic side effects is important for pancreatic cancer patients, who are typically diagnosed at an advanced stage. In this study, we found that the four-gene score was significantly associated with levels of drug sensitivity AUC of several agents in primary and metastatic pancreatic cancer cell lines. We cannot help but speculate that the four-gene score may have a role in patient selection for drug treatment for pancreatic cancer patients, in both the primary and metastatic settings. We believe our results warrant further study to analyze the correlation between the four-gene core and the treatment effect in future. In this study, we showed the utility of the score for pancreatic cancer under surgery. Considering that many pancreatic cancer patients are non-resectable, it may be more useful to apply this score to the pre-surgery sample, such as Endoscopic Ultrasound-Guided Fine Needle Aspiration and Biopsy (EUS-FNA/FNB) sample, like other studies [57] in the future.

This study still has some limitations. Although we have utilized multiple cohorts to validate these novel findings, it remains a retrospective study. Not all the cohorts had comprehensive clinical data, which may be a potential bias by the background. For example, clinical relevance of several gene expressions is known to vary by race and ethnic groups. It is very well-known that pancreatic cancer occurs at a higher rate in male African-Americans, and analyses of the cohort without them may mislead the efficacy of the four-gene score to that population. The validation of the score should be performed with cohorts with ethnic diversity that reflects the population in the future. The histological grade used in some cohorts was an older system; thus, it would have been ideal to present representative pathology. However, we were unable to do so because we downloaded the cohorts from the public domain, and did not have access to pathology data. Furthermore, treatment response studies were based on cell line data only, because it is difficult to obtain human sample data for assessing treatment response. It is known that the results of in vitro experiments can differ from the results in humans. A prospective study will be required in order to establish the four-gene score as a prognostic and predictive biomarker for these limitations. It is known that half of all pancreas cancer patients suffer recurrence even after R0 resection, which implies that the four-gene score does not reflect the behavior of micrometastasis, given that there was no survival difference by score after R0 resection. Finally, experimental studies are needed to fully verify and investigate the mechanism and applicability of our findings. In particular, the score may also be a useful tool for patient selection for chemotherapy in unresectable patients by using data in specimens collected by preoperative scrutiny.

## 4. Materials and Methods

### 4.1. Pancreatic Cancer Cohorts and Their Data

Clinical information and transcriptomic profiling of 176 pancreatic cancer patients in the TCGA pancreatic cancer cohort (TCGA_PAAD) [12], as a discovery cohort, was downloaded from the Genomic Data Commons Data Portal (GDC). We used the prior editions to the seventh edition of the American Joint Committee on Cancer (AJCC) for the clinical staging system. Alteration data was downloaded from cBioportal [58]. We also used published data of Hussain et al. (GSE62452; tumor sample; *n* = 69) [13], Chaika et al. (GSE34153; *n* = 75) [16], Chen et al. (GSE57495; *n* = 63) [14], and Jen et al. (GSE71729; *n* = 357) [15] to investigate the clinical relevance of the G2M pathway scores from the Gene Expression Omnibus (GEO) repository. Independent pancreatic ductal adenocarcinoma (PDAC) cohorts, GSE57495 and GSE62452, were used as validation cohorts. For cell line data, DepMap portal provide genetic and drug sensitivity AUC data (https://depmap.org/portal/). Details of the cell lines are shown in Appendix A. Normalized genomic data, which were transformed for log2, were used in all analysis. The average value was used for genes with multiple probes.

### 4.2. Gene Set Expression Analyses

For gene set enrichment analyses, gene set enrichment analyses (GSEA) software (Lava version 4.0) [59], and MSigDB Hallmark gene set collections [60] were used. And statistical significance was determined to a nominal *p* value threshold of 0.05 and a false discovery rate (FDR) of 0.25, as recommended by the GSEA software.

### 4.3. Cell Composition Fraction Estimation

The xCell algorithm was used to calculate xCell score through transcriptomic data [24] to analyze tumor immune microenvironment. The score was obtained through xCell website (https://xcell.ucsf.edu/), as we previously reported [61,62,63,64].

### 4.4. Statistical Analysis

For all data analysis and data plotting in the study, R software (version 4.0.1, R Project for Statistical Computing) and Microsoft Excel (version 16 for Windows) were used. Each four-gene score was calculated as (1.355 × (expression^DOK4^)) + (1.641 × (expression^HCCS^)) + (1.345 × (expression^PGF^)) + (1.232 × (expression^SHCBP1^)). Cytolytic activity score was calculated using the gene expression of granzyme A (*GZMA*) and perforin (*PRF1*) [65]. IFN-γ response score was from Thorsson et al. [66]. To divide low and high each pathway score groups, the median value of each score within cohorts was used. For comparison analysis between groups, statistical significance was determined to a *p* value less than 0.05 by one-way analysis of variance (ANOVA) and two-tail Fisher’s exact tests. Tukey-type boxplots showed median and interquartile level values.

## 5. Conclusions

We found that the four-gene score identified poor survival in pancreatic cancer, and has potential as a predictive biomarker for R0 resection and treatment response in metastatic pancreatic cancer. Evaluating the degree of the four-gene score could be a valuable prognostic and predictive tool in the future.

## Figures and Tables

**Figure 1 cancers-12-03635-f001:**
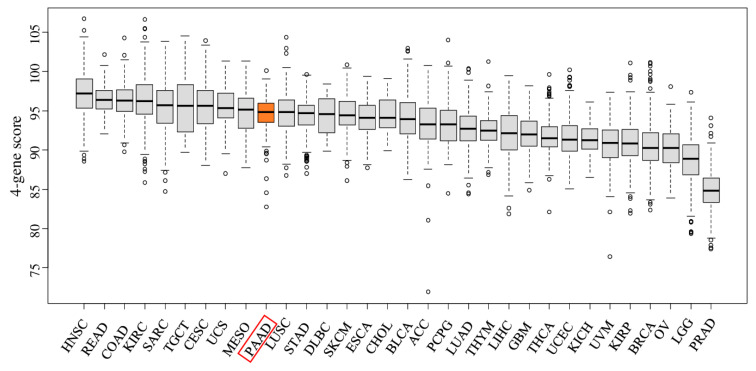
The four-gene score across multiple cancer types. Boxplots of the four-gene score comparing all cancer types from The Cancer Genome Atlas (TCGA) cohort. Tukey-type boxplots showed median and inter-quartile level values. ACC: adrenocortical carcinoma; BLCA: bladder urothelial carcinoma; BRCA: breast invasive carcinoma; CESC: cervical squamous cell carcinoma and endocervical adenocarcinoma; CHOL: cholangiocarcinoma; COAD: colon adenocarcinoma; DLBC: lymphoid neoplasm, diffuse large B-cell lymphoma; ESCA: esophageal carcinoma; GBM: glioblastoma multiforme; HNSC: head and neck squamous cell carcinoma; KICH: kidney chromophobe; KIRC: kidney renal clear-cell carcinoma; KIRP: kidney renal papillary cell carcinoma; LGG: brain lower-grade glioma; LIHC: liver hepatocellular carcinoma; LUAD: lung adenocarcinoma; LUSC: lung squamous cell carcinoma; MESO: mesothelioma; OV: ovarian serous cystadenocarcinoma; PAAD: pancreatic adenocarcinoma; PCPG: pheochromocytoma and paraganglioma; PRAD: prostate adenocarcinoma; READ: rectum adenocarcinoma; SARC: sarcoma; SKCM: skin cutaneous melanoma; STAD: stomach adenocarcinoma; TGCT: Testicular Germ Cell Tumors; THCA: thyroid carcinoma; THYM: thymoma; UCEC: uterine corpus endometrial carcinoma; UCS: uterine carcinosarcoma; UVM: uveal melanoma.

**Figure 2 cancers-12-03635-f002:**
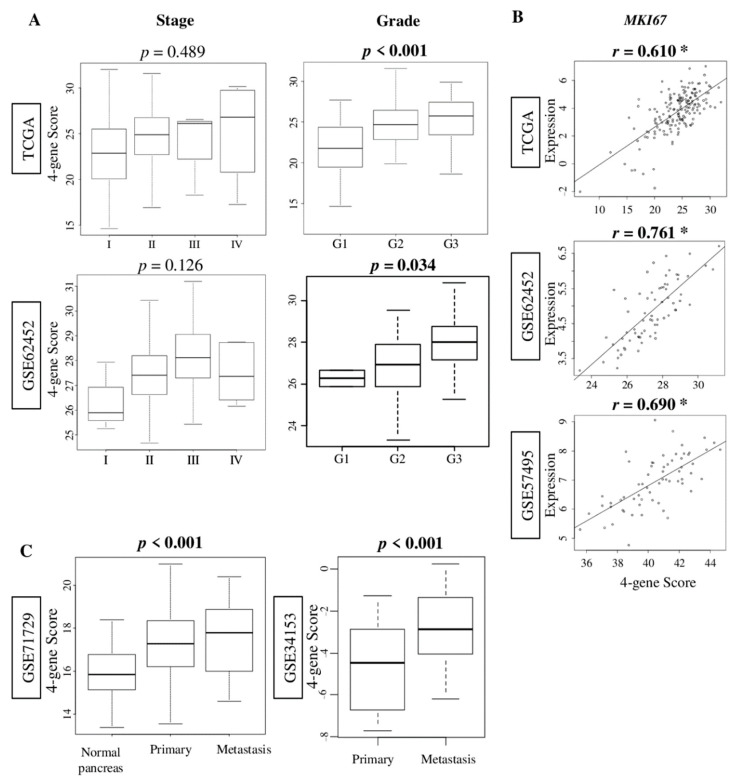
Association of the four-gene score with cell proliferation and metastatic pancreatic cancer. (**A**) Boxplots of the four-gene score by the American Joint Committee on Cancer stage, and Nottingham pathological grade in the TCGA, GSE62452, and GSE57495 cohorts. One-way ANOVA was used for the calculation. (**B**) Correlation plots of the four-gene score with expression of the *MKI67* gene in the TCGA, GSE62452, and GSE57495 cohorts. Spearman correlation statistics were used in the analysis. * *p*-value < 0.01. (**C**) Boxplot of the four-gene score between a normal pancreas, as well as primary and metastatic pancreatic cancer in GSE71729 and GSE34153 cohorts. One-way ANOVA was used to the calculation. Tukey-type boxplots showed median and interquartile level values.

**Figure 3 cancers-12-03635-f003:**
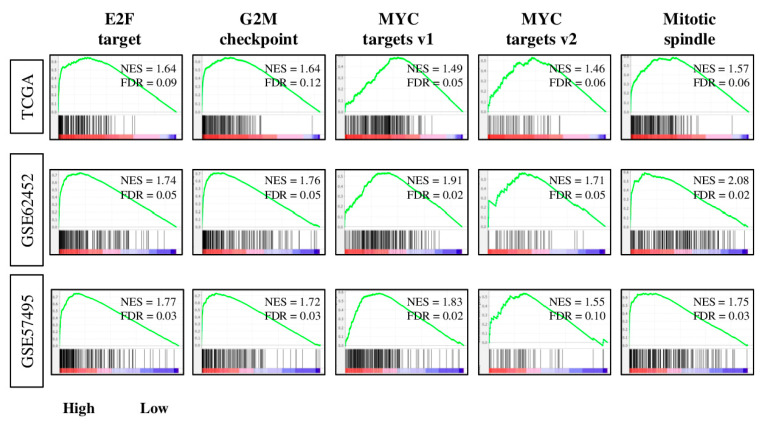
Gene set enrichment analysis (GSEA) of high-four-gene-score pancreatic cancer in the TCGA, GSE62452, and GSE57495 cohorts. Enrichment plots, along with normalized enrichment score (NES) and false discovery rate (FDR), are shown for cell proliferation-related Hallmark gene sets. The NES and FDR were determined with the classical GSEA method, where an FDR < 0.25 is considered significant.

**Figure 4 cancers-12-03635-f004:**
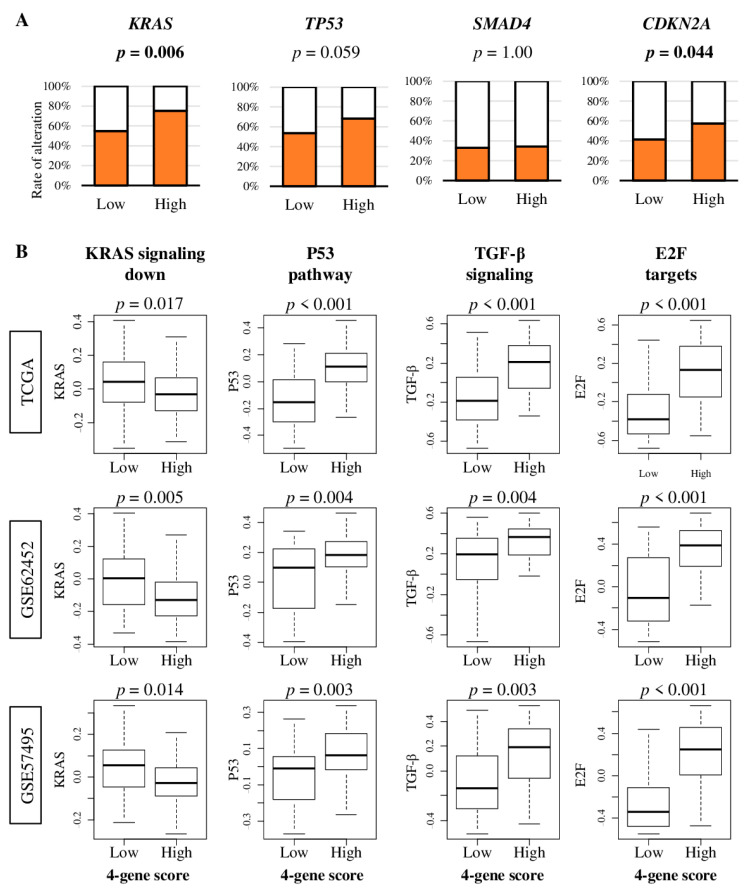
The association of the four-gene score with rate of *KRAS*, *TP53*, *SMAD4*, and *CDKN2A* gene alterations, and with signaling of *KRAS*, P53, transforming growth factor (TGF)-β, and E2F. (**A**) Bar plots of the percentage of patients with alteration of each genes in low- or high-four-gene-score groups. The Fisher’s exact test was used to calculate *p* values. (**B**) Boxplots of gene set variation analysis (GSVA) Hallmark gene sets, *KRAS* down-signaling, p53 pathway, TGF-β signaling, and E2F target by low- and high-four-gene-score group in the TCGA, GSE62452, and GSE57495 cohorts. Low- and high-four-gene score groups were divided by the median value of each cohort. A one-way ANOVA test was used to calculate *p* values. Tukey-type boxplots showed median and interquartile level values.

**Figure 5 cancers-12-03635-f005:**
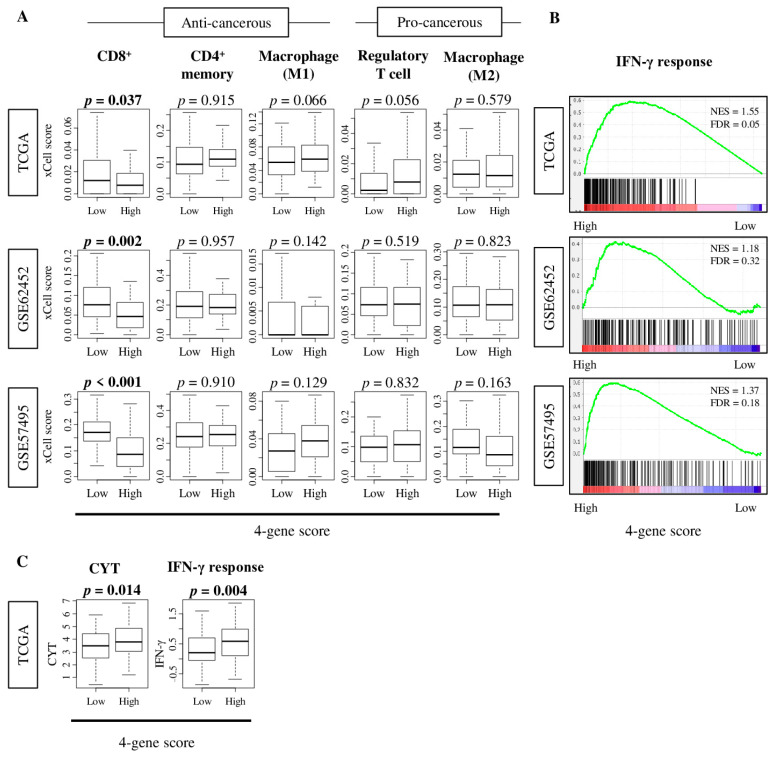
The association of the four-gene score with tumor-infiltrating immune cells and immune function scores in the TCGA, GSE62452, and GSE57495 cohorts. (**A**) Boxplots of the xCell scores of anti-cancerous immune cells—CD8^+^ T cells, CD4^+^ memory T cells, M1 macrophages, and pro-cancerous immune cells—regulatory T cells, and M2 macrophages, by low- and high-four-gene scores in three cohorts. A one-way ANOVA test was used to calculate *p* values. (**B**) Enrichment plot of the interferon (IFN)-γ response pathway of Hallmark gene sets using GSEA in three cohorts. NES and FDR were determined with the classical GSEA method, where FDR < 0.25 was considered significant. (**C**) Boxplots of the cytolytic activity score (CYT) and IFN-γ response score by low- and high-four-gene scores in the TCGA cohort. A one-way ANOVA test was used to calculate *p* values. Tukey-type boxplots showed median and interquartile level values.

**Figure 6 cancers-12-03635-f006:**
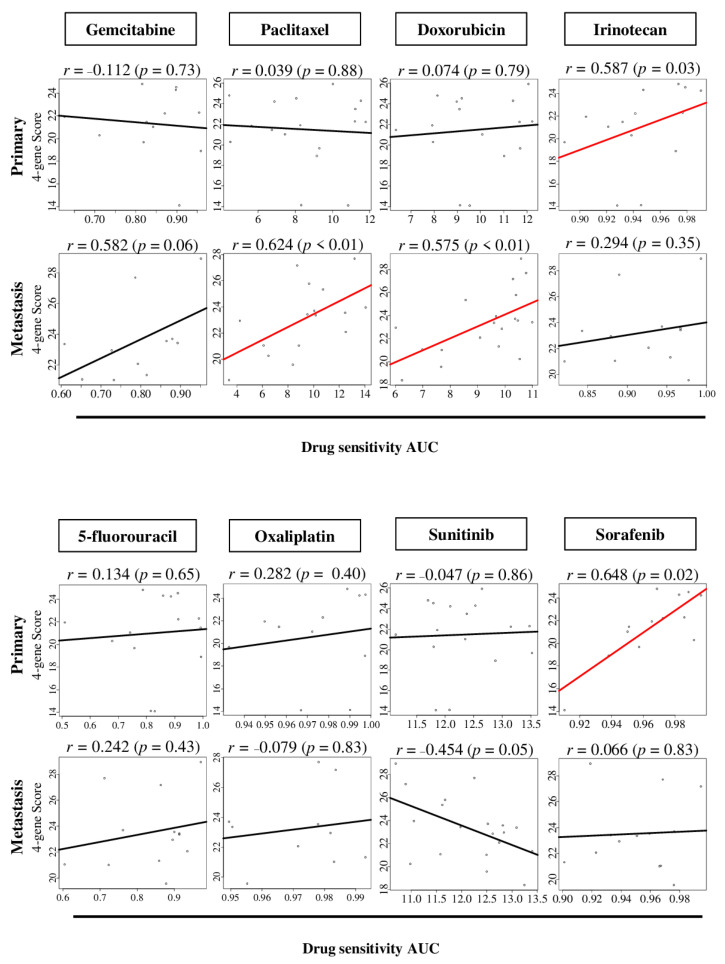
The correlation of the four-gene score with drug sensitivity of several drugs, from cell line data from the DepMap portal. The correlation plots are between the four-gene score and the level of drug sensitivity area under curve (AUC) of gemcitabine, paclitaxel, doxorubicin, irinotecan, 5-fluorouracil, oxaliplatin, sunitinib, and sorafenib in primary and metastatic pancreatic cancer cell lines. Spearman correlation statistics were used for the analysis.

**Figure 7 cancers-12-03635-f007:**
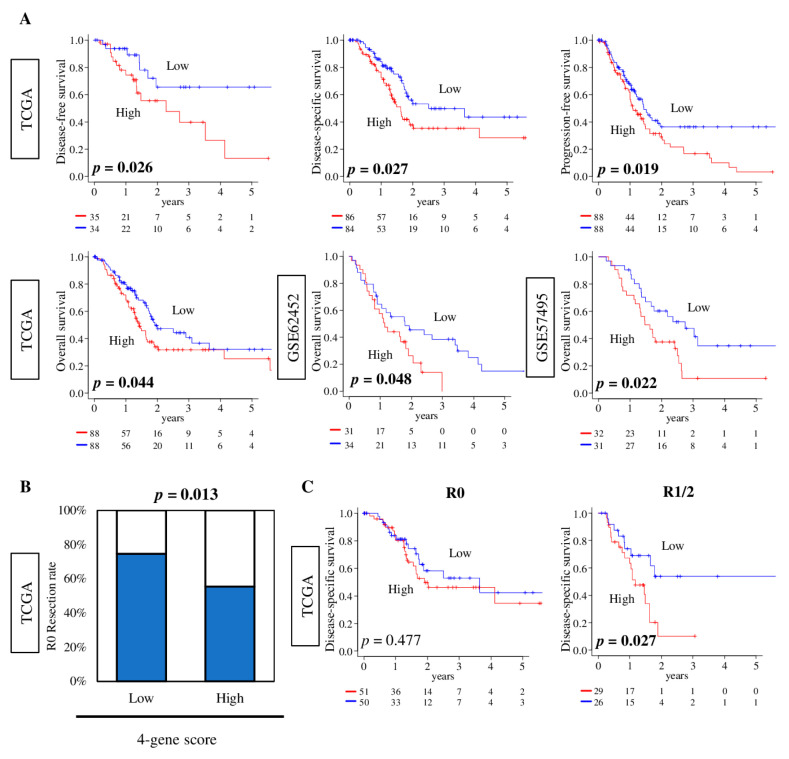
The association of the four-gene score with survival of pancreatic cancer patients and resection status. (**A**) Kaplan–Meier survival plots of comparison between low (blue line) and high (red line) four-gene score groups for disease-free survival (DFS), disease-specific survival (DSS), and progression-free survival (PFS) in the TCGA cohort, and overall survival (OS) in the TCGA, GSE62452, and GSE57495 cohorts. A log–rank test was used to calculate the *p* values. (**B**) Bar plot of achievement rate of R0 resection (blue) between low and high four-gene scores in the TCGA cohort. Fisher’s exact test was used to calculate the *p* value. (**C**) Kaplan–Meier survival plots of comparison between low- and high-four-gene-score groups, in an R0 or R1/2 resection, for DSS in the TCGA cohort. A log–rank test was used to calculate the *p* values.

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
