# Peer review of "A Novel Four-Gene Score to Predict Pathologically Complete (R0) Resection and Survival in Pancreatic Cancer"

_cancers, 2020, doi:10.3390/cancers12123635_

Round 1

Reviewer 1 Report

Pancreatic cancer is one of the most aggressive cancers with a very low five year survival rate after treatment. Discovery of biomarkers to aid diagnosis, prediction of virulence, and optimal therapeutics of patients would be a major breakthrough in the clinical setting. The current study is the first step to achieve these goals. The current study is based on successful prior research results on a four-gene score as a prognostic and predictive biomarker panel for breast cancer, which was identified using bioinformatics across publically available datasets. This study extends that prior to another prevalent, but more deadly cancer, pancreatic cancer.

The methodology in this manuscript is solid because is utilizes established bioinformatics pipeline created from prior research in breast cancer, which was referenced prominently throughout the manuscript, from summary and abstract to introduction and discussion. The novelty is in application, not in the bioinformatics techniques,  per se. Acknowledge the novelty in the current manuscript accurately, otherwise the risk is to thwart acceptance of the current research and undermines the importance of previous research results from the prior breast cancer study.

  • Excellently written

Minor comments to be addressed in the manuscripts before publication:

  • Refrain from calling the methods in this paper novel because you clearly reference your work in breast cancer repeatedly.
  • Highlight that the application of the methodology is novel to pancreatic cancer (PC) in the introduction and discussion
  • Provide examples of the pancreatic histological staging of PC pathology used because an noncurrent, older staging system was used to grade the samples. Examples will strength the acceptance of the data and facilitate adoption of the 4-gene panel in pancreatic cancer. Without examples in the manuscript, most clinicians won’t be able to adopt it in their practice.
  • It is misleading to repeatedly state “954 patients in total were used in the study (for example lines 33-36). Please add “….954 patient were used for both discovery and validation of the 4-gene score from publicly available datasets for pancreatic cancer” in the different sections
  • Most importantly, add references to discussion section about the clinical attributes regarding race, ethnicity, and sex, of all the datasets used for both the discovery and validation of the 4-gene score. It is very well known that pancreatic cancer occurs at a higher rate in male African Americans but nothing was mentioned regarding the patient race, ethnicity, and sex in the discussion. Omission of these facts and risk factors is not acceptable because your 4-gene score may be applicable to a limited population of patients due to limitations in the diversity in the cohorts used to establish and validate the 4-gene score. A similar finding has been found for African Americans and Alzheimer’s Disease, which lack an association with a known genetic risk factor, APOE variants, found in other races. Consequently, this is critical because the 4-gene score may be used in patients that will not benefit, but these patients may receive harm applying a 4-gene score that was developed using the genomes of other specific ethnic and racial cohorts.

Reviewer 2 Report

The authors investigated the clinical usefulness of a  4 gene signature score that has been previously reported to be associated with prognosis in breast cancer, in PDAC cohorts.

The study is for the vast majority based on an "in silico" analysis of data from publicly available datasets.

The authors found that this score is associated with prognosis of resected PDAC and correlates with a number of proliferation markers and with common mutations in PDAC (kras, SMAD4, p53, CDKN2A). Notably, there is also an association with the immune infiltrate.

Finally, in the only part of the study that is not "in silico" they investigated the association between the score and the response to different drugs of PDAC cell lines, with findings of an association with treatment with paclitaxel and sorafenib.

Overall the study is well written and easy to be followed.

My main concerns regard the actual clinical significance and the lack of original experiments and validation.

I would advice strongly:

  1. To test the score on an independent set of samples of PDAC patients.
  2. Alternatively the authors may investigate whether some of the 4 targets have a significance beyond their prognostic role by silencing them in cell lines or animal models (PKC).
  3. The authors insist on the relevance of the score to predict prognosis after surgery. However, most PDAC patients will never get resection. I believe that the expression of these targets may be relevant if avallable on EUS-FNA/FNB samples as this would allow a molecular classiifcation of patients at diagnosis before surgery (cite PMID: 31798775). This should be discussed. 
  4. Some of the figures can be supplementary material. 
  5. I can't see details of the cell lines that have been employed for the drug sensitivity analyses. Most PDAC cell lines are well characterized for their mutational status, hence an analysis of the association of common mutations and the score is possible and should be performed.

Round 2

Reviewer 2 Report

The manuscript has improved and limitations are underlined.